# Vitamin D Deficiency Does Not Impair Diastolic Function in Elite Athletes

**DOI:** 10.3390/medicina61030407

**Published:** 2025-02-26

**Authors:** Ömer Özkan, İdris Yakut, Gürhan Dönmez, Feza Korkusuz

**Affiliations:** 1Gaziler Physical Therapy and Rehabilitation Training and Research Hospital, 06800 Ankara, Turkey; 2Sincan Training and Research Hospital, 06949 Ankara, Turkey; idrislive@windowslive.com; 3Faculty of Medicine, Department of Sports Medicine, Hacettepe University, 06100 Ankara, Turkey; gdonmez_1805@yahoo.com (G.D.); feza.korkusuz@gmail.com (F.K.)

**Keywords:** athlete, sport, echocardiography, athlete’s heart, diastolic, 25(OH)D

## Abstract

*Background and Objectives*: Regular exercise is known to induce cardiovascular adaptations collectively referred to as “athlete’s heart”. While previous research has explored the morphological and functional cardiac adaptations in athletes, the relationship between vitamin D (25-hydroxyvitamin D [25(OH)D]) levels and echocardiographic parameters remains underexplored. This study aims to assess the association between 25(OH)D levels and structural and functional cardiac parameters using electrocardiographic (ECG) and echocardiographic evaluations in athletes. *Materials and Methods*: This case–control study included 93 male athletes, categorized into professional (*n* = 68) and recreational (*n* = 25) groups. Professional athletes were further divided into football (*n* = 19), weightlifting (*n* = 22), and running (*n* = 27) subgroups. Serum 25(OH)D levels were measured using high-performance liquid chromatography–tandem mass spectrometry (LC-MS/MS). Standard 12-lead ECG and transthoracic echocardiography were performed to assess cardiac structure and function. Data were analyzed using statistical tests that were appropriate for normal and non-normal distributions, with a significance level set at *p* < 0.05. *Results*: Athletes exhibited higher left ventricular interventricular septum (IVS) thickness and left ventricular posterior wall thickness (LVPWd) compared to the control group. Significant differences in diastolic function parameters, including early (E) and late (A) diastolic filling velocities and the E/A ratio, were observed among athlete subgroups. The weightlifting group showed lower end-systolic diameter (ESD) values than the football group. However, no statistically significant relationship was found between 25(OH)D levels and echocardiographic diastolic parameters. While more than half of the athletes had insufficient 25(OH)D levels (<30 ng/mL), their average values were higher than those reported in previous studies. *Conclusions*: This study demonstrates that 25(OH)D levels do not significantly influence echocardiographic diastolic parameters in athletes. However, notable differences in structural and functional cardiac findings were observed among different sports disciplines. These findings contribute to the understanding of cardiac adaptations in athletes and suggest that 25(OH)D may not play a crucial role in diastolic function. Further research is needed to explore the long-term effects of vitamin D on athletic cardiac performance.

## 1. Introduction

The cardiovascular benefits of regular exercise and physical activity are well-established currently [1,2]. Physical activity helps regulate blood pressure, controls blood lipid levels, and increases insulin sensitivity. Health authorities and current guidelines recommend at least 150 min of moderate exercise per week [3]. Both professional and recreational athletes often train for significantly more than these recommended levels. In addition to its numerous metabolic benefits, intense training induces specific cardiovascular adaptations, collectively known as athlete’s heart [4]. An athlete’s heart and its responses to various factors are crucial for athletic participation and performance.

Cardiac adaptation occurs through two primary types of exercise: dynamic and static. In athletes performing dynamic exercise, a slight increase in arterial pressure is observed, alongside an increase in heart rate (HR) and cardiac output (CO) reaching up to 40 L per minute [5]. In these athletes, eccentric hypertrophy is expected [6]; whereas in static exercise athletes, such as weightlifters performing short-duration, high-intensity exercises, an increase in arterial pressure along with a slight increase in HR and CO is observed. In weightlifters, arterial pressure can rise to as high as 480/350 mmHg during training [7], which corresponds to concentric hypertrophy in static exercises [6]. The clinical evaluation of the heart’s response to exercise, known as “athlete’s heart”, assessed electrophysiologically and echocardiographically, has recently become a subject of significant interest. It is well-known that exercise causes various adaptations, such as sinus bradycardia, and increases in IVS and LVPWd values [8].

Vitamin D, also named 25-hydroxyvitamin D [25(OH)D], is an essential nutrient involved in multiple physiological processes, including calcium metabolism, bone health, immune function, and the regulation of cell growth and differentiation. 25(OH)D deficiency has been suggested as a potential risk factor for cardiovascular diseases. Considering the importance of cardiovascular function in athletic performance, 25(OH)D levels may also be associated with an athlete’s cardiovascular capacity, potentially influencing endurance, oxygen utilization, and overall physical performance [9]. Some researchers indicate that vitamin D deficiency may adversely affect cardiac structure and function. Vitamin D receptors are present in cardiac myocytes and fibroblasts, suggesting a direct role in myocardial health. Studies have shown that severe vitamin D deficiency in athletes is associated with smaller cardiac dimensions, including reduced left ventricular mass and smaller aortic root diameters [10]. These structural alterations could potentially impair cardiac output and athletic performance. Furthermore, vitamin D deficiency has been linked to increased arterial stiffness and endothelial dysfunction, factors that may compromise cardiovascular efficiency during intense physical activity [11]. Despite these findings, the specific impact of vitamin D levels on cardiac adaptations in athletes remains underexplored, warranting further investigation into this relationship.

However, the number of studies investigating the relationship between 25(OH)D levels, cardiac structure, and function in athletes using echocardiographic evaluation is limited. Additionally, studies examining systolic and diastolic functions, along with 25(OH)D and related blood parameters, within the athlete population are still scarce. The objective of this study is to reveal the relationship between the levels of 25(OH)D and the structural and functional values of the heart assessed through electrocardiographic and echocardiographic parameters.

## 2. Method

### 2.1. Participants

This research was designed as a case–control study. Among the participants, 14 were excluded from the study due to reasons such as failure to complete the examinations or not attending ongoing tests. Athletes who had taken vitamin D supplements within the last 12 months were not included to the study. All blood samples for 25(OH)D measurements were collected during the summer months (July–August, 2019). Thus, a total of 93 male athletes were included in the study. The participants consisted of professional athletes (*n* = 68) and recreational athletes (control group, *n* = 25). Professional athletes were divided into three subgroups: football (*n* = 19), weightlifting (*n* = 22), and running (*n* = 27) (Figure 1). The football players included in the study were licensed professional athletes. The weightlifters consisted of athletes at the national-team level. Long-distance runners with a personal best half-marathon time of 1 h and 12 min were included in the running group. Recreational athletes (control group) consisted of individuals who regularly engaged in aerobic training at least 2–3 times per week, primarily through running. Their exercise routines typically consisted of moderate-intensity activities performed for general health and fitness rather than performance optimization. Ethical committee approval was received from the Ethics Committee of Hacettepe University (Approval No: 19-692).

### 2.2. Electrocardiograms and 25(OH)D Status Assessment

Blood tests and electrocardiography examinations were evaluated after 12 h of fasting. Serum levels of 25(OH)D were measured. Serum was separated by centrifugation, and 25(OH)D levels were assessed using high-performance liquid chromatography–tandem mass spectrometry (LC-MS/MS) with the Shimadzu LCMS-8040 (Shimadzu Corp., Kyoto, Japan), following standardized laboratory operating procedures. 25(OH)D levels were classified as deficiency (<20.0 ng/mL) and suboptimal/optimal (≥20.0 ng/mL). A standard 12-lead ECG was obtained after a 5 min rest in the supine position. At the end of the recordings, R in aVL, Cornell values, and heart rates were calculated from the electrocardiograms.

### 2.3. Echocardiography

Echocardiographic evaluation was performed on all athletes by an experienced cardiologist. All evaluations were conducted transthoracically using an EpiQ-7 (Koninklijke Philips N.V., Eindhoven, The Netherlands) echocardiography device. Imaging of the heart was performed in standard planes. Participants were placed in the left lateral decubitus position, and measurements were taken using 2D, M-mode, Color Doppler, and flow Doppler in parasternal long and short axes, apical four-chamber, and five-chamber views. Suprasternal and subcostal evaluations were performed in the supine position. In line with the current consensus, evaluations were conducted midday to account for circadian rhythm effects on cardiac diastolic functions. All evaluations were conducted by a single experienced cardiologist. Measurements of left ventricular dimensions, such as interventricular septum thickness (IVS), left ventricular posterior wall thickness (LVPWd), end-systolic diameter (EDD), and end-diastolic diameter (ESD), were evaluated in the parasternal long axis. Left ventricular end-systolic and end-diastolic volumes were measured using the modified Simpson method, and EF values were calculated accordingly. Right ventricular systolic function was examined using tricuspid annular plane systolic excursion (TAPSE) in M-mode evaluations. Left ventricular mass (LVM) calculations were based on the Devereux formula [12]. Left atrial volume was calculated using the ‘prolate’ ellipsoid method [13]. Left ventricular diastolic function was evaluated using Tissue Doppler measurements of early filling velocity (E), late filling velocity (A), the E/A ratio, deceleration time (DT), early-diastolic mitral annulus velocity (e’) (average of septal and lateral values), and the ratio of E to e’ (E/e’).

### 2.4. Statistical Analyses

Statistical analyses were performed using SPSS software (version 24.0, IBM, Armonk, NY, USA). The normality of the distribution for numerical variables was assessed using visual methods (histograms and probability plots) and analytical tests (Kolmogorov–Smirnov/Shapiro–Wilk tests). Descriptive statistics were presented as mean ± standard deviation for normally distributed numerical variables, and percentages (%) for categorical variables. Chi-square analyses were conducted to assess differences in sun exposure time according to sports disciplines. For comparisons, Student’s *t*-test and one-way ANOVA were used for normally distributed numerical variables, whereas the Mann–Whitney U test and Kruskal–Wallis test were applied for non-normally distributed variables. For multiple comparisons following ANOVA, Tukey’s post hoc test was used to determine pairwise differences between groups. Differences in echocardiographic parameters among groups based on 25(OH)D levels were analyzed using ANCOVA. A *p*-value of less than 0.05 was considered statistically significant in all comparisons.

## 3. Results

### 3.1. Sociodemographic Data

The data were obtained from 93 elite athletes. A total of 93 participants were divided into four groups: football (*n* = 19), weightlifting (*n* = 22), running (*n* = 27), and a control group (*n* = 25). (Table 1) The mean ages for the football, weightlifting, running, and control groups were 21.5 ± 3.8, 19.1 ± 1.0, 30.4 ± 6.1, and 28.3 ± 8.3 years, respectively. No significant differences were observed in SBP between the groups. There was significant sports-type differences in the anthropometric indices of the BMI (body mass index) as shown in Table 1. The ages of the running and control groups were significantly higher than those of the football and weightlifting groups, with notable differences between the following pairs: control vs. football (*p* < 0.001), control vs. weightlifting (*p* < 0.001), football vs. running (*p* < 0.001), and weightlifting vs. running (*p* < 0.001). Regarding the BMI, the weightlifting and control groups exhibited higher average values compared to the football and running groups. Significant differences were noted between the following pairs: control vs. football (*p* < 0.001), control vs. running (*p* < 0.001), football vs. weightlifting (*p* = 0.005), and weightlifting vs. running (*p* = 0.001). 25(OH)D levels of the athletes were as follows: football players’ levels were 25.1 ± 6.1 ng/mL, weightlifters’ levels were 19.9 ± 5.9 ng/mL, runners’ levels were 24.9 ± 7.5 ng/mL, and the control group’s levels were 22.2 ± 9.0 ng/mL. Statistical differences in 25(OH)D levels were identified between the weightlifting and running groups (*p* = 0.028) and between the weightlifting and football groups (*p* = 0.04). Among the IPAQ (The International Physical Activity Questionnaire) scores, the weightlifting group had the highest values, while the football group had higher scores than the other groups. The IPAQ scores of the participants varied across sports disciplines. Weightlifters had an average score of 5589.7 ± 2621.6, while runners exhibited the highest activity levels with 8580.2 ± 1926.8. Football players recorded a mean score of 3631.4 ± 1691.5, and the control group had the lowest activity levels, averaging 3093.6 ± 2372.2. There were significant differences between the control and football groups (*p* = 0.001), control and weightlifting groups (*p* < 0.001), football and weightlifting groups (*p* = 0.001), and football and running groups (*p* < 0.001). Sun exposure duration varied significantly among groups (*p* < 0.05). The highest proportion of athletes with more than 120 min of daily sun exposure was observed in football players (63.2%), followed by runners (14.8%), weightlifters (13.6%), and the control group (36%). A sun exposure duration of 60–120 min was most common among runners (40.7%), whereas 30–60 min of exposure was predominantly seen in weightlifters (63.6%) and the control group (44%). Additionally, three weightlifters (13.6%) and two control participants (8%) reported no sun exposure. The significant difference between groups was primarily due to higher sun exposure in football players and runners compared to weightlifters and the control group (*p* < 0.05).

### 3.2. Athletes’ ECG Measurements

No statistically significant differences were observed between groups in terms of resting heart rate or R wave height in aVL derivation (R in aVL) on ECGs. However, there was a statistically significant difference in Cornell voltage values (*p* < 0.05), primarily attributed to the difference between the running and control groups (*p* = 0.005) (Table 1).

### 3.3. Structural Ventricular Echocardiographic Parameters

The left ventricular interventricular septum (IVS) thickness was higher in athlete groups, with significant differences observed between the control group and both the football and running groups (*p* = 0.022 vs. control, *p* = 0.028 vs. control, respectively). For left ventricular posterior wall thickness (LVPWd), a significant difference was found between the football and weightlifting subgroups (*p* = 0.002). Regarding end-systolic diameter (ESD), a statistically significant difference was noted between the football and weightlifting groups. (*p* = 0.002) No significant differences were observed for other parameters.

### 3.4. Echocardiographic Diastolic Assessments of Athletes

A significant difference in early diastolic filling velocity (E) was found among the groups (Table 2). Higher average E values were observed in athlete subgroups, and the difference was statistically significant between the weightlifting and running groups (*p* = 0.012). For late diastolic filling velocity (A), a significant difference was attributed to the football and running groups (*p* = 0.047). Similarly, for the E/A ratio, a statistically significant difference was found between the football and running groups (*p* = 0.036). No significant differences were observed for E/e’, e’ septal, or e’ lateral values between the groups. Although the DT value was higher in athlete groups, the differences were not statistically significant. No significant differences were observed for other parameters.

### 3.5. Relationship Between Echocardiographic Diastolic Parameters and 25(OH)D

Echocardiographic diastolic parameters were assessed by categorizing groups based on 25(OH)D levels that were above or below 20 ng/mL, as shown in Table 3. A statistically significant difference was found between the athlete and control groups in terms of the E value, with higher values observed in the athlete group compared to the control group (*p* = 0.03). No statistically significant differences were observed for other groups and parameters.

## 4. Discussion

This study examined the cardiac response to intense exercise and recovery, as well as their relationship with 25(OH)D across different sports disciplines categorized based on their dynamic and static components. While most sports involve a combination of both, the groups in this study were categorized as follows: high dynamic–low static (running), moderate dynamic–moderate static (football, mixed type), and low dynamic–high static (weightlifting) [14].

In our study, there was a difference between athlete groups in terms of age and BMI and IPAQ scores. Given that aerobic training reduces the BMI and that similar differences have been observed in previous echocardiography studies involving athletes, this result was expected [15,16]. All athletes, except for the control group, had been performing at a high level in their respective sports for at least 7 years. From this perspective, regardless of the age difference between groups, cardiac adaptation would be expected to have occurred.

When 25(OH)D levels were evaluated with the threshold of 30 ng/mL considered as sufficient, more than half of the athletes had insufficient 25(OH)D levels. However, when compared with the literature on 25(OH)D levels in athletes, the average levels in our study could be considered high [17,18,19]. This could be related to the athletes’ professionalism, even though no supplements were used. In elite athletes, it has been reported that the difference may be due to outdoor sports activities and good nutritional habits, compared to the general population [20].

Upon examining the ECG values of the athletes, a significant difference in Cornell values between the athlete and control groups was found. The running group showed higher Cornell values compared to the control group. This difference is expected in terms of physiological hypertrophy in athlete hearts. Despite this, none of the athletes had Cornell values exceeding the upper limit of 24 mV for men.

In our study, athletes showed higher IVS values compared to the control group. As for LVPWd values, although the athletes generally showed higher values, the weightlifting group was different from the other athlete subgroups. When subgroup comparisons were made, the football group’s LVPWd values were higher than those of the weightlifting group. These findings were consistent with studies that evaluated the morphological characteristics of the heart in trained athletes [6,21,22,23]. The separation of the weightlifting subgroup from the general athlete group can be explained by their lower average age and the longer time required for morphological changes in the heart. Longer training durations may be necessary for changes in LCPWd and IVS in the weightlifting discipline [24]. Also, professional athletes did not differ statistically from the recreational athlete group, which may be attributed to the presence of similar athlete’s heart adaptations observed in recreational runners, comparable to those seen in professional athletes.

When comparing ESD and EDD, the weightlifting group showed lower ESD values compared to the football group. No significant differences were found in these two parameters in other pairwise comparisons. A study by Silva et al. found no difference in ESD and EDD between weightlifters and long-distance runners [25]. Another study reported no significant changes in ESD and EDD after 8 weeks of resistance training [26]. Some studies in the athlete population have reported high ESD and EDD values [16,27], while others found no differences [28,29]. In a study by Moro et al., significant differences in ESD and EDD were found between cyclists and football players compared to the control group, but no significant difference was found for the running group [28]. Our findings suggest that while exercise had a more pronounced effect on interventricular septum and posterior wall dimensions, it did not result in significant changes in ESD and EDD [29].

When evaluating ventricular diastolic function, an important parameter is the E/A ratio. In our study, no differences in the E/A ratio were found among the football, weightlifting, and control groups. However, the running group had significantly lower E/A values compared to the football group. Previous studies have shown that the E/A ratio in athletes is either normal or slightly increased compared to sedentary control groups [5,30,31]. However, some studies have shown that athletes’ groups do not significantly differ from control groups [32,33]. In our study, the weightlifting group showed higher E values compared to the running group, whereas the running group had higher A values compared to the football group. The differences in these values, within the normal range of the E/A ratio, may not be meaningful since the E/A ratio is also influenced by heart rate, preload, and afterload. Kneffel et al. demonstrated that increased E/A values are associated with reduced cardiac output and sinus bradycardia as an adaptive mechanism in athletes [34]. Although there was a negative correlation between these two values in our study (r = −0.19, *p* = 0.068), it was not statistically significant. All athletes in the study, except for one athlete in the running group (E/A: 0.95), had E/A values above 1.

In our study, no statistically significant relationship was found between 25(OH)D levels and diastolic echocardiographic parameters. The relationship between 25(OH)D levels and diastolic parameters in the heart has been previously studied in different patient groups, and conflicting results have been reported in the literature. A study by Sonkar et al. found no relationship between 25(OH)D levels and diastolic parameters in non-diabetic adults with chronic kidney disease [35]. Similarly, in a retrospective study by Pandit et al. including 1011 patients, no relationship was found between 25(OH)D levels and the LA filling time index, E/e’, e, LVM, and DT [36]. In contrast to our study and the mentioned studies, another study of 34 children with chronic kidney disease found a correlation between parathyroid hormone and 25(OH)D levels with E’ and E/E’ values. While previous studies have investigated the relationship between 25(OH)D levels and the heart’s morphological and systolic functions in athletes, this relationship has not been extensively studied in athletes [10]. Therefore, our study is important in this aspect. While observational studies suggest a link between low vitamin D levels and increased cardiovascular disease (CVD) risk, clinical trials have not confirmed significant benefits of supplementation. A large study of 25,871 participants found no reduction in major cardiovascular events with daily vitamin D supplementation (2000 IU) over 5.3 years [37]. Similarly, the Women’s Health Initiative and the DIMENSION study found no improvements in cardiovascular outcomes or endothelial function [38]. Meta-analyses, including 21 randomized controlled trials, have shown no significant impact of vitamin D on major adverse cardiovascular events, stroke, myocardial infarction, or cardiovascular mortality [39]. While low vitamin D levels are associated with higher CVD risk, supplementation has not been proven to prevent heart disease or improve cardiac function. Future studies should explore potential benefits in specific populations. The effects of vitamin D on athletic performance and cardiac adaptations remain uncertain, with studies producing mixed results. While some research suggests that vitamin D supplementation may improve aerobic capacity and strength, other studies report no significant effects on sprint speed, inflammation, or bone health [40]. Owens et al. also highlight conflicting findings regarding vitamin D’s role in cardiovascular adaptations, with some studies linking deficiency to impaired cardiac function, while others find no direct benefit of supplementation [41]. Additionally, randomized trials in both general and athletic populations fail to demonstrate consistent improvements in cardiovascular performance. Research suggests that while severe vitamin D deficiency can impact cardiac structure, moderate insufficiency may not significantly affect function, as athletes with low vitamin D levels have shown no impairments in myocardial performance [10]. Additionally, high-intensity training enhances cardiovascular efficiency, potentially compensating for lower vitamin D levels, with studies showing that well-trained athletes can maintain optimal cardiac performance regardless of vitamin D variations [42]. Furthermore, once a sufficient vitamin D level is reached, additional increases do not necessarily improve myocardial function, reinforcing the idea that vitamin D effects may be negligible beyond a threshold [36]. These inconsistencies suggest that vitamin D supplementation should be considered on an individual basis, with further research needed to determine its true impact on athletes.

Because 25(OH)D analysis and echocardiographic assessments were performed only once, potential seasonal or temporal variations in 25(OH)D levels and their positive or negative effects on the heart may have been overlooked. Long-term follow-up studies with 25(OH)D supplementation could be conducted to address this. Another limitation of our study is the lack of nutritional status analysis, which may influence vitamin D levels and cardiac function. However, the selection of the athletes for the study and the fact that the tests and evaluations were conducted in the same season strengthen our study. The use of a recreational sports group as the control group was chosen to exclude confounding factors that might arise between the athlete and control groups. However, the absence of a sedentary healthy group could be considered a limitation. The presence of such a group might have made the differences in systolic and diastolic parameters more pronounced. Nevertheless, we believe that the choice of a recreational sports group as the control group helped exclude potential confounding factors. Increasing the number of participants could further strengthen the study. However, the aim of revealing the characteristics of elite-level athletes limited the number of athletes included. Future studies could include larger groups of athletes and control groups. Similarly, there were differences in demographic and anthropometric data among athlete groups, which stemmed from the limited population of the study. Since there are studies in the literature showing the relationship between VO2max and cardiac adaptation, VO2max could have been used as an objective parameter for classifying athletes. The ratios of morphological data to body surface area were not calculated for all parameters. Comparisons of morphological and systolic function values, made without considering body surface area, may be less valuable in this regard.

## 5. Conclusions

In conclusion, our study examined the relationship between 25(OH)D levels and echocardiographic parameters, which had not been explored in the literature before, and while no significant relationship was found, higher 25(OH) levels were observed in athlete groups compared to the control group. Additionally, higher ECG and structural and functional echocardiographic values were found in the athlete groups. This study highlights the differences in structural and functional cardiac findings among athletes in various sports and provides valuable information on the effects of dynamic and static exercise types, offering a basis for future studies in similar populations. This study has demonstrated that there is no relationship between 25(OH)D and echocardiographic diastolic parameters in athletes.

## Figures and Tables

**Figure 1 medicina-61-00407-f001:**
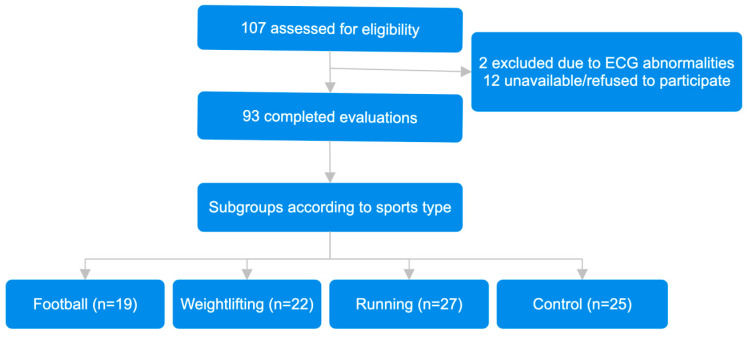
Study patient flow diagram.

**Table 1 medicina-61-00407-t001:** Baseline characteristics of study participants.

*N* = 93 Mean ± SD		Football (*n* = 19)	Weightlifting (*n* = 22)	Running (*n* = 27)	Control (*n* = 25)	*p*	η^2^
Age		21.5 ± 3.8 ^a,b^	19.1 ± 1.0 ^a,b^	30.4 ± 6.1	28.3 ± 8.3	**<0.05**	0.42
BMI (kg/m^2^)		21.7 ± 1.9 ^a,c^	25.1 ± 5.1	21.8 ± 1.4 ^a,c^	25.4 ± 3.3	**<0.05**	0.23
SBP (mmHg)		116.6 ± 8.8	118.2 ± 10.6	114.5 ± 7.5	117.9 ± 7.4	0.3	0.33
25(OH)D (ng/mL)		25.1 ± 6.1 ^c^	18.9 ± 5.9	24.9 ± 7.5 ^c^	22.2 ± 9.0	**<0.05**	0.11
IPAQ score-total		5589 ± 2621 ^a,b,c^	8580 ± 1926 ^a^	3631 ± 1691	3093 ± 2372	**<0.05**	0.51
Sun exposure time (min/day)	<30	0 (0%)	3 (13.6%)	0 (0%)	2 (8%)	**<0.05**	
	30–60	5 (26.3%)	14 (63.6%)	12 (44.4%)	11 (44%)		
	60–120	2(10.5%)	2 (9.2%)	11(40.7%)	3(12%)		
	<120	12 (63.2%)	3 (13.6%)	4 (14.8%)	9 (36%)		
Electrocardiographic values							
R in aVL (mm)		2.64 ± 2.05	3.15 ± 1.70	3.40 ± 2.25	2.75 ±2.04	0.6	0.02
Cornell (mm)		11.41 ± 5.33	10.31 ± 3.91	13.36 ± 4.77 ^a^	8.50 ± 4.27	**<0.05**	0.05

BMI: body mass index; SBP: systolic blood pressure; 25(OH)D: Vitamin D; R in aVL: R wave in aVL lead. IPAQ score: International Physical Activity Questionnaire; min: minute. ^a^ Significant difference compared to control group from post hoc test. ^b^ Significant difference compared to running group from post hoc test. ^c^ Significant difference compared to weightlifting group from post hoc test.

**Table 2 medicina-61-00407-t002:** Echocardiographic parameters of athletes and control groups.

*N* = 93	Football (*n* = 19)	Weightlifting (*n* = 22)	Running (*n* = 27)	Control (*n* = 25)	*p*	η^2^
Echocardiographic structural values						
IVS (mm)	0.99 ± 0.12 ^a^	0.95 ± 0.12	0.99 ± 0.11 ^a^	0.92 ± 0.11	**<0.05**	0.08
LVPWd (mm)	1.02 ± 0.91 ^c^	0.91 ± 0.11	0.99 ± 0.11	0.93 ± 0.10	**<0.05**	0.14
EDD (mm)	4.61 ± 0.41	4.54 ± 0.31	4.62 ± 0.35	4.60 ± 0.41	0.8	0.01
ESD (mm)	3.03 ± 0.37 ^c^	2.71 ± 0.33	2.91 ± 0.41	3.12 ± 0.38	**<0.05**	0.15
LVM (g)	157.8 ± 30.2	142.4 ± 34.9	153.2 ± 27.9	140.2 ± 33.1	0.2	0.05
EF (%)	65.5 ± 2.3	65.2 ± 2.3	65.4 ± 2.4	69.9 ± 15.0	0.9	0.01
CO (L/min)	4.2 ± 1.2	4.6 ± 1.6	4.1 ± 0.9	4.2 ± 1.1	0.8	0.03
TAPSE (cm)	2.7 ± 0.4	2.8 ± 0.4	2.8 ± 0.4	2.6 ± 0.5	0.6	0.02
LAV (ml/m^2^)	28.7 ± 7.3	26.6 ± 5.9	29.6 ± 4.7	30.1 ± 8.6	0.3	0.04
Echocardiographic diastolic values						
E (cm/s)	88.4 ± 14.3	88.9 ± 14.5 ^b^	79.5 ± 13.7	78.1 ± 16.3	**<0.05**	0.10
A (cm/s)	46.2 ± 6.0 ^b^	51.1 ± 10.0	52.1 ± 8.4	46.2 ± 9.5	**<0.05**	0.09
E/A	1.9 ± 0.4 ^b^	1.8 ± 0.5	1.6 ± 0.4	1.8 ± 0.6	**<0.05**	0.09
E/e’	6.3 ± 1.2	6.1 ± 0.8	5.7 ± 1.4	5.7 ± 0.9	0.3	0.04
DT (ms)	183.2 ± 53.2	153.9 ± 25.5	161.7 ± 21.5	153.4 ± 47.2	0.2	0.08
e’ septal (cm/s)	12.7 ± 1.8	12.6 ± 1.8	12.3 ± 2.5	12.4 ± 2.3	0.9	0.01
e’ lateral (cm/s)	15.9 ± 2.9	16.1 ± 3.3	15.3 ± 2.7	15.2 ± 3.1	0.7	0.02

IVS: interventricular septum thickness; LVPWd: left ventricular posterior wall thickness; EDD: end-diastolic diameter; ESD: end-systolic diameter; LVM: left ventricular mass; EF: ejection fraction; CO: cardiac output; TAPSE: tricuspid annular plane systolic excursion; LAV: left atrial volume; E: early filling velocity; A: late filling velocity; DT: deceleration time; e’: early diastolic mitral annulus velocity (average of septal and lateral values). ^a^ Significant difference compared to control group from post hoc test. ^b^ Significant difference compared to running group from post hoc test. ^c^ Significant difference compared to weightlifting group from post hoc test.

**Table 3 medicina-61-00407-t003:** Covariance analysis of the relationship between athletes’ echocardiographic diastolic values and 25(OH)D.

25(OH)D		E ^a^	A	E/A	E/e’	DZ	e’ Septal	e’ Lateral
Athlete (*n* = 68)	>20 ng/mL	83.3 ± 13.8	49.9 ± 8.0	1.7 ± 0.4	5.9 ± 1.2	164.4 ± 36.4	12.6 ± 1.9	15.5 ± 2.6
	<20 ng/mL	88.6 ± 16.0	50.6 ± 10.0	1.8 ± 0.5	6.2 ± 1.0	166.9 ± 35.4	12.2 ± 2.4	16.3 ± 3.7
Control (*n* = 25)	>20 ng/mL	78.7 ± 17.3	44.6 ± 8.2	1.9 ± 0.7	5.7 ± 0.8	155.9 ± 46.7	12.2 ± 2.4	15.5 ± 3.7
	<20 ng/mL	77.4 ± 15.9	47.9 ± 10.9	1.7 ± 0.4	5.7 ± 0.9	150.7 ± 49.8	12.6 ± 2.3	14.8 ± 2.4
*p*		0.4	0.5	0.09	0.7	0.7	0.4	0.3
ηp2		0.009	0.004	0.03	0.002	0.002	0.007	0.013

25(OH)D: vitamin D; E: early filling velocity; A: late filling velocity; DT: deceleration time; e’: early diastolic mitral annulus velocity (average of septal and lateral values). ηp2: partial eta squared. ^a^ Significant difference between athlete vs. control group (*p* < 0.05) (F = 4.745) (ηp2 = 0.05).

## Data Availability

The data presented in this study are available on request from the corresponding author. Due to privacy and ethical considerations, the data cannot be shared publicly. Personal sensitive data is protected under the Law No. 6698 on the Protection of Personal Data, published in 2016. Access to the data may be granted upon reasonable request, subject to compliance with legal and ethical regulations.

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
