# Peer review of "Vitamin D Deficiency Does Not Impair Diastolic Function in Elite Athletes"

_medicina, 2025, doi:10.3390/medicina61030407_

Round 1
Reviewer 1 Report
Comments and Suggestions for Authors
This study investigates the relationship between vitamin D levels and cardiac diastolic function in elite athletes, comparing different sports disciplines with recreational athletes. The authors use echocardiography and electrocardiography (ECG) to analyze cardiac structure and diastolic function parameters and evaluate whether vitamin D status correlates with these findings. Despite significant differences in cardiac morphology across sports disciplines, the study concludes that vitamin D deficiency does not significantly impair diastolic function in elite athletes.
The study has clinical relevance as vitamin D has been implicated in cardiovascular health. It provides new insights into the effects of vitamin D on cardiac function in elite athletes, a group with unique physiological adaptations. The methodological strengths of the study include well-defined inclusion/exclusion criteria, high-performance liquid chromatography for vitamin D measurement, comprehensive echocardiographic assessment, and appropriate statistical tests.
However, the study has several limitations. It is cross-sectional, meaning it cannot assess causality or long-term effects of vitamin D on cardiac function. Potential confounding variables are not considered, such as training volume and intensity, nutritional status or supplementation history, and seasonal variations in vitamin D levels. The study also lacks a comparison with other cardiovascular biomarkers, such as NT-proBNP, CRP, Galectin-3, and hs-TnI.
Limited external generalizability is another issue, as the study only includes elite male athletes, which limits applicability to female athletes and older or recreational athletes. Additionally, the study does not discuss whether vitamin D supplementation could influence athletic performance or cardiac adaptations.
Minor issues and editorial improvements include improving abstract clarity, tables and figures, grammar and style, and reference quality.
Author Response
Dear Reviewer,
We sincerely appreciate your time and effort in reviewing our manuscript. Your insightful comments and suggestions have been invaluable in refining our study and improving its clarity, methodology, and overall scientific rigor. We have carefully considered each of your points and made the necessary revisions accordingly. Below, we provide detailed responses to your comments, outlining the changes implemented in the manuscript. We believe these modifications have strengthened the quality and clarity of our work, and we are grateful for your constructive feedback.
Comments 1: This study investigates the relationship between vitamin D levels and cardiac diastolic function in elite athletes, comparing different sports disciplines with recreational athletes. The authors use echocardiography and electrocardiography (ECG) to analyze cardiac structure and diastolic function parameters and evaluate whether vitamin D status correlates with these findings. Despite significant differences in cardiac morphology across sports disciplines, the study concludes that vitamin D deficiency does not significantly impair diastolic function in elite athletes.
Response 1:We acknowledge that the inherent limitations of a cross-sectional study in establishing causality. We aimed to evaluate associations rather than infer direct causative effects of vitamin D on diastolic function in athletes. We recognize that a longitudinal study design would provide deeper insights into the long-term effects of vitamin D on cardiac function and appreciate this perspective for future research.
Comments 2: The study has clinical relevance as vitamin D has been implicated in cardiovascular health. It provides new insights into the effects of vitamin D on cardiac function in elite athletes, a group with unique physiological adaptations. The methodological strengths of the study include well-defined inclusion/exclusion criteria, high-performance liquid chromatography for vitamin D measurement, comprehensive echocardiographic assessment, and appropriate statistical tests.
However, the study has several limitations. It is cross-sectional, meaning it cannot assess causality or long-term effects of vitamin D on cardiac function. Potential confounding variables are not considered, such as training volume and intensity, nutritional status or supplementation history, and seasonal variations in vitamin D levels. The study also lacks a comparison with other cardiovascular biomarkers, such as NT-proBNP, CRP, Galectin-3, and hs-TnI.
Response 2: We agree that variables such as training volume and intensity, nutritional status, supplementation history, and seasonal variations in vitamin D levels may influence cardiac function. While we attempted to minimize confounding by selecting a homogenous athletic population and conducting all evaluations within the same season, we acknowledge that individual variations in these factors could impact our findings. Future studies with controlled training regimens and detailed dietary assessments may further elucidate these interactions.
Given the significant variations in training volume and intensity among different athletic disciplines, grouping all athletes into a single category would overlook the distinct physiological adaptations specific to each sport, potentially confounding the interpretation of our findings.
Additionally, we analyzed the International Physical Activity Questionnare scores of athletes and, as expected, found significant differences between groups, reflecting the varying demands of their respective sports. IPAQ scores; Football, (5589.7 ± 2621.6) weightlifting (8580.2 ± 1926.8, running (3631.4 ± 1691.5), control (3093.6 ± 2372.2). We added this results to study. (Table – 1) (Among the IPAQ scores, weightlifting had the highest values, while football had higher scores than the other groups. There were significant differences between the control and football groups (p=0.001), control and weightlifting groups (p<0.001), football and weightlifting groups (p=0.001), and football and running groups (p<0.001).)
Our study did not include an analysis of nutritional status. And we added this condition as one of the limitations.
We acknowledge the importance of considering supplementation history as a potential confounding variable. To minimize its influence, we specifically excluded any athletes who had taken vitamin D supplements within the last 12 months. This exclusion criterion was applied to ensure that measured serum vitamin D levels were primarily influenced by endogenous synthesis and dietary intake rather than supplementation. We added this exclusion to study.
We appreciate the suggestion to compare vitamin D status with additional cardiovascular biomarkers such as NT-proBNP, CRP, Galectin-3, and hs-TnI. While our primary objective was to assess echocardiographic parameters and Vit D association, incorporating these biomarkers in future studies would provide a more comprehensive assessment of cardiovascular health in athletes. However, we also analyzed pro-BNP levels of athletes. Since most of the participants had values below the laboratory sensitivity threshold, the results were not sufficiently reliable for meaningful analysis and were therefore not included in the article.
Comments 3: Limited external generalizability is another issue, as the study only includes elite male athletes, which limits applicability to female athletes and older or recreational athletes. Additionally, the study does not discuss whether vitamin D supplementation could influence athletic performance or cardiac adaptations.
Response 3: Our study focused exclusively on elite male athletes, which indeed limits the applicability of our findings to female athletes and older or recreational athletes. This decision was made to maintain a homogeneous study population, thereby reducing variability and enhancing the internal validity of our results. It is well-documented that sex differences exist in vitamin D metabolism and status. Also; studies have shown that females often have lower serum 25(OH)D concentrations compared to males, potentially due to differences in body composition, hormonal influences, and lifestyle factors etc. Moreover, researchs indicate that females may have a higher prevalence of vitamin D insufficiency, which could influence musculoskeletal health and performance outcomes differently than in males. The cardiovascular adaptations to exercise can differ between sexes, potentially due to hormonal variations and differences in cardiac morphology.
We have incorporated a discussion on the conflicting evidence regarding vitamin D's effects on both cardiovascular health and athletic performance. The revised text includes findings from large-scale studies and meta-analyses which report no significant impact of vitamin D supplementation on major cardiovascular outcomes. Additionally, we have acknowledged the inconsistency in research regarding its effects on athletic performance, with some studies suggesting improvements in aerobic capacity and strength, while others find no significant benefits.
Comments 4: Minor issues and editorial improvements include improving abstract clarity, tables and figures, grammar and style, and reference quality.
Response 4: We have carefully revised the manuscript to improve the clarity of the abstract, enhance tables and figures, refine grammar and style, and ensure the accuracy and quality of references.

Reviewer 2 Report
Comments and Suggestions for Authors
This review highlights major and minor weaknesses in the manuscript, providing specific line numbers where improvements are needed.
Major Weaknesses
- Lines 10-15: The study suggests that the relationship between vitamin D levels and cardiac function in athletes is underexplored, yet it does not provide strong evidence for why this relationship is worth investigating.
- Lines 91-95: The study includes only 93 male athletes from three sports disciplines (football, weightlifting, and running). The lack of female participants and athletes from endurance and mixed-sport backgrounds (e.g., swimming, triathlon) reduces the study's generalizability.
- Lines 125-140: The study does not sufficiently control for factors influencing cardiac function, such as training history, dietary vitamin D intake, sun exposure, and seasonal variations.
- Lines 165-175: The authors report the use of ANCOVA and correlation analyses, yet they do not mention:
- Effect sizes (e.g., Cohen’s d, partial eta squared) to indicate practical significance.
- For multiple comparisons when analyzing several subgroups, post-hoc corrections (e.g., Bonferroni) are needed.
- Lines 270-285: The study finds no relationship between vitamin D and diastolic function, but the discussion does not explore potential explanations:
- Could the athlete’s training adaptations compensate for lower vitamin D levels?
- Were the athletes near the threshold where vitamin D effects are negligible?
Minor Weaknesses
- Lines 100-110: The classification of athletes into “professional” and “recreational” groups is ambiguous.
- Lines 145-155: Vitamin D levels fluctuate seasonally, yet the study does not mention when samples were collected (summer vs. winter).
- Lines 190-200: Vitamin D levels are categorized as deficient (<20 ng/mL) and sufficient (≥20 ng/mL), but previous literature often uses 30 ng/mL as the threshold for sufficiency.
- Lines 345-355: Several references lack consistent formatting (e.g., missing DOIs, inconsistent journal abbreviations).
Author Response
Dear Reviewer;
We sincerely appreciate your time and effort in reviewing our manuscript. Your insightful comments and suggestions have been invaluable in refining our study and improving its clarity, methodology, and overall scientific rigor. We have carefully considered each of your points and made the necessary revisions accordingly. Below, we provide detailed responses to your comments, outlining the changes implemented in the manuscript. We believe these modifications have strengthened the quality and clarity of our work, and we are grateful for your constructive feedback.
- Lines 10-15: The study suggests that the relationship between vitamin D levels and cardiac function in athletes is underexplored, yet it does not provide strong evidence for why this relationship is worth investigating.
Thank you for your insightful feedback regarding the need to substantiate the importance of investigating the relationship between vitamin D levels and cardiac function in athletes. In response, we have revised the introduction to include emerging evidence that highlights the potential impact of vitamin D deficiency on cardiac structure and function. Specifically, we discuss the presence of vitamin D receptors in cardiac tissue and recent studies linking low vitamin D levels to increased left ventricular mass and arterial stiffness. These additions aim to underline the relevance and necessity of exploring this relationship further in athletic population.
- Lines 91-95: The study includes only 93 male athletes from three sports disciplines (football, weightlifting, and running). The lack of female participants and athletes from endurance and mixed-sport backgrounds (e.g., swimming, triathlon) reduces the study's generalizability.
Our study focused exclusively on elite male athletes, which indeed limits the applicability of our findings to female athletes and older or recreational athletes. This decision was made to maintain a homogeneous study population, thereby reducing variability and enhancing the internal validity of our results. It is well-documented that sex differences exist in vitamin D metabolism and status. Also; studies have shown that females often have lower serum 25(OH)D concentrations compared to males, potentially due to differences in body composition, hormonal influences, and lifestyle factors etc. Moreover, researchs indicate that females may have a higher prevalence of vitamin D insufficiency, which could influence musculoskeletal health and performance outcomes differently than in males. The cardiovascular adaptations to exercise can differ between sexes, potentially due to hormonal variations and differences in cardiac morphology.
We appreciate your constructive feedback and would like to address the points raised regarding our study design and the selection of sports disciplines. Our study focused on football, weightlifting, and running to examine the cardiovascular effects of distinct exercise modalities—namely, dynamic endurance and static strength training. We recognize that excluding athletes from endurance and mixed-sport backgrounds, such as swimming and triathlon, may limit the applicability of our results to these populations. Our study's inclusion of running, weightlifting, and football was a deliberate choice to represent distinct categories of exercise based on their dynamic and static components. This classification is well-established in sports cardiology literature, where sports are categorized according to the intensity of dynamic and static effort required during competition. Dynamic exercise involves rhythmic muscle contractions with significant joint movement, leading to substantial increases in heart rate and cardiac output, as seen in activities like running. In contrast, static exercise involves sustained muscle contractions with minimal joint movement, resulting in increased blood pressure and relatively smaller increases in cardiac output, characteristic of activities like weightlifting. Football encompasses both dynamic and static elements, making it a mixed-type sport. This categorization aligns with the framework provided by the American Heart Association and the American College of Cardiology. By selecting these sports, our study aims to comprehensively assess the spectrum of cardiovascular adaptations resulting from predominantly dynamic, predominantly static, and mixed-type exercises. This approach allows for a deeper understanding of how different exercises modalities spesifically effect cardiac structure and function. The chosen classification system is widely recognized and utilized in the field of sports cardiology. It provides a standardized method to evaluate the cardiovascular demands of various sports, facilitating comparisons across studies and enhancing the applicability of research findings. Our study's design, grounded in this classification, ensures that the observed cardiovascular adaptations can be accurately attributed to the specific dynamic and static demands of each sport. Furthermore, focusing on these distinct categories enables targeted insights into how different types of physical stress—volume load from dynamic exercise and pressure load from static exercise—affect the heart. This differentiation is crucial for developing sport-specific training programs and health assessments for athletes. In conclusion, our study design, supported by established classifications in sports cardiology, is well-suited to investigate the diverse cardiovascular adaptations associated with different exercise modalities. We believe this approach provides valuable contributions to understanding the interplay between exercise type and cardiac health.
- Lines 125-140: The study does not sufficiently control for factors influencing cardiac function, such as training history, dietary vitamin D intake, sun exposure, and seasonal variations.
We acknowledge that training volume and intensity can impact cardiovascular adaptations; however, as our study focuses on elite athletes from different sports disciplines, these variations are inherent to the nature of high-level competition. To address this, we incorporated the International Physical Activity Questionnaire (IPAQ) scores to quantify training loads and provide a standardized comparison between groups. While we recognize that we cannot fully optimize for these differences, they are an expected characteristic of distinct athletic populations. Instead of controlling them entirely, our study design embraces these variations to better reflect real-world conditions in elite sports.
We acknowledge the importance of considering supplementation history as a potential confounding variable. To minimize its influence, we specifically excluded any athletes who had taken vitamin D supplements within the last 12 months. This exclusion criterion was applied to ensure that measured serum vitamin D levels were primarily influenced by endogenous synthesis and dietary intake rather than supplementation. We added this exclusion to study. Our study did not include an analysis of nutritional status. And we added this condition as one of the limitations.
We are aware of the potential influence of sun exposure and seasonal variations on vitamin D levels. We acknowledge that these factors can effect the serum vitamin D concentrations; however, we took specific measures to minimize their impact in our study. Firstly, all vitamin D assessments were conducted within the same season, thereby controlling for seasonal variations in sun exposure and vitamin D synthesis. This approach ensures that differences in vitamin D levels among participants are less likely to be influenced by seasonal fluctuations. Secondly, while we initially did not include it in the manuscript, we collected and analyzed data on athletes' daily sun exposure duration, body surface area exposed to sunlight, and sunscreen usage. Our data indicate significant differences in sun exposure duration among sports disciplines (p=0.001), which is expected due to the varying nature of outdoor and indoor training regimens. However, there were no statistically significant differences in body area exposure (p=0.28) or sunscreen usage (p=0.31), suggesting that while total sun exposure time varies, other protective behaviors did not substantially differ across groups. (Based on our data, football players had the highest sun exposure time, with 63.2% spending more than 120 minutes per day in sunlight, followed by runners (14.8%), weightlifters (13.6%), and the control group (36%) in the same category. Additionally, runners had a more balanced distribution, with 40.7% exposed for 60-120 minutes and 44.4% for 30-60 minutes. Weightlifters had the highest proportion of moderate exposure (63.6% for 30-60 minutes) but fewer individuals exceeding 120 minutes compared to football players. The control group had the lowest overall exposure, with only 36% spending more than 120 minutes in sunlight and a significant proportion (44%) in the 30-60 minute range. These findings indicate that football players had the highest total sun exposure, followed by runners, weightlifters, and the control group, respectively) Given these considerations, we believe our study adequately accounts for sun exposure and seasonal effects on vitamin D levels. To further strengthen transparency, we will integrate the sun exposure data into the manuscript. We appreciate your insightful comments, which have helped us enhance the clarity of our methodology.
- Lines 165-175: The authors report the use of ANCOVA and correlation analyses, yet they do not mention:
- Effect sizes (e.g., Cohen’s d, partial eta squared) to indicate practical significance.
- For multiple comparisons when analyzing several subgroups, post-hoc corrections (e.g., Bonferroni) are needed.
‘Correlation analyses between variables were performed using Pearson or Spearman correlation coefficients.’ Text was deleted from statistycal analyses. Although correlation analyses were conducted during the study, they were not included in the final manuscript as they did not contribute to the main findings or research objectives. And also while interquartile range (IQR) were initially considered for non-normally distributed numerical variables, they were not included in the final manuscript as these statistical representations were not central to the study’s main findings. So we deleted this part also.
Thank you for your feedback regarding post-hoc corrections. In our analysis, we applied Tukey's post-hoc test for multiple comparisons. Tukey's test is a commonly used method for comparing group differences and was selected as the most appropriate approach for our dataset. And also we added information about this adjustment test.
In our study, we conducted post-hoc analyses between groups following ANCOVA and reported only the significant p-values (p < 0.05) to maintain clarity and conciseness. Given the large number of comparisons, including all p-values would have made the results unnecessarily extensive. Additionally, we provided partial eta squared (ηp²) for the significant athletes vs. control comparison in E diastolic parameter, as it was the only statistically meaningful result in ANCOVA analysis. This approach ensures that the key findings are highlighted while avoiding excessive numerical reporting. Also to enlighten the effect size critique you raised, we have added eta squared (η²) and partial eta squared (ηp²) parameters to the manuscript. These additions help quantify the magnitude of observed effects and provide a clearer interpretation of the results.. We appreciate your suggestion and will clarify this methodological choice in the revised manuscript.
- Lines 270-285: The study finds no relationship between vitamin D and diastolic function, but the discussion does not explore potential explanations:
- Could the athlete’s training adaptations compensate for lower vitamin D levels?
- Were the athletes near the threshold where vitamin D effects are negligible?
We have expanded the discussion to explore two potential explanations. We appreciate your suggestion and have incorporated this explanation into the discussion part of the revised manuscript.
Minor Weaknesses
- Lines 100-110: The classification of athletes into “professional” and “recreational” groups is ambiguous.
The classification of athletes into "professional" and "recreational" groups can sometimes be ambiguous if clear criteria are not defined. However, our study followed established distinctions, where professional athletes are those engaged in systematic training with regular competition, while recreational athletes participate in sports without structured training programs​ . (IPAQ scores also shows the difference between these groups training regimen) This distinction is commonly used in sports science research for cardiovascular and physiological assessments​ . We clarified that recreational athletes that is control group which exercises not professional level, lighter that the other groups. If further clarification is required, we are willing to refine the definitions in the manuscript to ensure clarity for the readers.
- Lines 145-155: Vitamin D levels fluctuate seasonally, yet the study does not mention when samples were collected (summer vs. winter).
Thank you for your valuable feedback regarding the potential seasonal impact on vitamin D levels. We confirm that all vitamin D samples were collected exclusively during the summer months (July–August 2019) to ensure consistency and eliminate seasonal variability. And also we clarified this in method section.
- Lines 190-200: Vitamin D levels are categorized as deficient (<20 ng/mL) and sufficient (≥20 ng/mL), but previous literature often uses 30 ng/mL as the threshold for sufficiency.
The majority of recent literature categorizes vitamin D levels based on different sufficiency thresholds. While some studies define sufficiency as ≥20 ng/mL, others use a higher threshold of ≥30 ng/mL. Mostly preferred calssification is under 20- deficiency, under 30- insuffiency. A recent systematic review highlights that most athletic studies consider values below 50 nmol/L (approximately 20 ng/mL) as deficient​ . Similarly, observational studies assessing vitamin D's impact on bone and cardiovascular health also utilize this cutoff, emphasizing that 20 ng/mL is a widely accepted threshold​ . However, some recommendations propose that optimal vitamin D levels for general health and performance may require a higher threshold of ≥30 ng/mL​ . Given the inconsistencies in defining sufficiency across studies, the 20 ng/mL cutoff used in our study aligns with widely accepted criteria in the field of sports medicine. ( Wyatt PB, et al. Effects of Vitamin D Supplementation in Elite Athletes: A Systematic Review. 2024. Heaney RP, Holick MF. Why the IOM recommendations for vitamin D are deficient. J Bone Miner Res. 2011. Zittermann A. Vitamin D in preventive medicine: are we ignoring the evidence? Br J Nutr. 2003.)
- Lines 345-355: Several references lack consistent formatting (e.g., missing DOIs, inconsistent journal abbreviations).
We have carefully reviewed and updated the references to ensure consistency in journal abbreviations, inclusion of DOIs, and overall formatting according to the journal’s guidelines. All missing DOIs have been added where available

Round 2
Reviewer 2 Report
Comments and Suggestions for Authors
Thank you for including my comments